# Relationship between Everyday Discrimination and Substance Use among Adolescents in Northern Chile

**DOI:** 10.3390/ijerph18126485

**Published:** 2021-06-16

**Authors:** Alejandra Caqueo-Urízar, Alfonso Urzúa, Patricio Mena-Chamorro, Jerome Flores, Matías Irarrázaval, Ellen Graniffo, David R. Williams

**Affiliations:** 1Instituto de Alta Investigación, Universidad de Tarapacá, Arica 1000000, Chile; 2Escuela de Psicología, Universidad Católica del Norte, Antofagasta 1240000, Chile; alurzua@ucn.cl; 3Temuco & Centro Justicia Educacional, Departamento de Psicología, Universidad de la Frontera, CJE, Santiago 7820436, Chile; pmena@uta.cl; 4Escuela de Psicología y Filosofía, Universidad de Tarapacá & Centro Justicia Educacional, CJE, Santiago 7820436, Chile; jflores@uta.cl; 5Departamento de Psiquiatría, Facultad de Medicina, Hospital Clínico, Universidad de Chile & Institute for Depression and Personality Research, MIDAP, Santiago 8380453, Chile; mirarrazavald@uchile.cl; 6Facultad de Educación y Humanidades, Universidad de Tarapacá, Arica 1000000, Chile; graniffoellen@gmail.com; 7Department of Social and Behavioral Sciences, Harvard T.H. Chan School of Public Health, Boston, MA 02115-5810, USA; dwilliam@hsph.harvard.edu; 8Department of African and African American Studies, Harvard University, Boston, MA 02115-5810, USA

**Keywords:** substance use, everyday discrimination, Chilean adolescents

## Abstract

Substance use is a public health problem that affects the normal physical, neurological, and psychological development of adolescents. Apparently, discrimination is an important variable for explaining the initiation and continued use of alcohol and marijuana. Since most research focused on discrimination based on factors, such as race, ethnicity, sexual orientation, or gender faced by minority groups, studies on discrimination faced by the general population remain scarce. This cross-sectional study described the relationship between everyday discrimination and alcohol and marijuana use-related behaviors among Chilean adolescents. It included 2330 students between 12 and 20 years of age from educational establishments in the city of Arica. To evaluate substance use, specifically alcohol and marijuana, the Child and Adolescent Evaluation System (SENA) was used. The Everyday Discrimination scale was used to evaluate discrimination. Age and everyday discrimination can predict up to 11% of the variance in substance use. Reducing the incidence of everyday discrimination may help reduce heavy alcohol and marijuana consumption among adolescents.

## 1. Introduction

The consumption of marijuana and alcohol is related to personal and psychosocial factors. As part of the latter, daily experiences of discrimination may become a potent risk factor for such substance use [1].

Substance use is considered a public health problem that affects the normal functioning of millions of people, especially young people, who are at a stage of continuous physical, neurological, and psychological development [2]. Research suggests that young people are more likely to experience the harmful effects of substance use than adults [2,3,4]. These entail increased risk of affective and anxiety disorders, violence, psychotic symptoms or disorders, fatal and non-fatal overdose, substance abuse, cognitive impairment, polysubstance use, traffic accidents, self-harm, poor academic performance, and early dropouts [5,6,7,8,9].

The negative consequences and possible long-term effects in adulthood have also emerged as a topic of research interest [10,11], and global trends highlight the importance of this issue. Alcohol consumption by adolescents worldwide is 26.5% and 5.6% for marijuana consumption [12,13], with Europe and Latin America being the regions with the highest rates of substance use.

In Chile, the 12th national study of drugs among the school population, conducted by the National Service for the Prevention and Rehabilitation of Drug and Alcohol Consumption [14], found that on an average, regardless of gender, adolescents tried alcohol and marijuana for the first time before the age of 15 years (13.7 years for alcohol and 14.4 years for marijuana). Additionally, alcohol consumption within this population was 31.1%, and marijuana consumption was 30.9%. Trends indicate that as students grow older, they increasingly use these two substances. The law in Chile permits the possession and use of “small” amounts of marijuana for “personal use”; its use under medical prescription is permitted, although it is available mainly through informal markets and home cultivation [15,16].

The initiation and maintenance of substance use in adolescents are characterized by fluctuations between use and cessation, with some young people maintaining moderate use for decades and never increasing the use, while others may exhibit intermittent cessation periods, permanently abstain, or increase rapidly, facilitating the development of substance use disorders [10,17].

The literature has focused on identifying which risk factors could have an influence on the onset and maintenance of substance use in adolescents and has documented that personality traits [18,19], neurological development [20], stress reactivity [21], influences of family, school, and peers [22,23,24]; access to alcohol and drugs [25], poverty [26], social exclusion, inequality, and discrimination [27,28] are the risk factors.

There is a growing research interest in the potential role of discrimination, which falls under the last factor mentioned above. Discrimination is a multifaceted and multidimensional phenomenon, which, according to Jones [29], manifests itself through four sub-types of discrimination: (1) individuals—understood as the daily actions of a personal and degrading nature that promote beliefs of inferiority among people; (2) cultural, which refers to the beliefs of superiority held by those belonging to the dominant group over the subordinate group; (3) institutionalized, which is characterized by differential access to opportunities in society based on systematic discrimination on the basis of race; and (4) collective, which manifests through the restriction or denial of basic rights and privileges of members of the minority group by the dominant group [1,30].

According to the minority stress model [31], discrimination is considered a biased behavior that favors the ingroup while causing harm or disadvantage to the outgroup, especially to members of the minority group who experience high levels of stress that are related to their minority status. More recently, from an integral perspective, the social resistance model proposed that the discrimination that minorities face can encourage members of these groups to actively participate (consciously or unconsciously) in diverse behaviors of daily resistance to express their opposition toward the majority group [32].

These daily resistance behaviors are mostly characterized by unhealthy behaviors that allow individuals to reduce stress and express their dissatisfaction. Specifically, a lack of attachment to the dominant culture, along with experiences of discrimination, may be associated with maladaptive responses that include excessive use of alcohol and marijuana [33]. Jones’ [29] conceptualization of individual discrimination, together with models of minority stress and social resistance, converges because discrimination is observed daily between different groups of individuals and produces stress in the affected group. Thus, any experience of chronic, routine discrimination or unfair treatment can be considered everyday discrimination.

Prior research on the relationship between discrimination and substance use suggests that, with greater experience of discrimination, the use of substances—including alcohol and marijuana—tends to increase [1]. A study on Caribbean adolescents in the United States demonstrated a positive association between discrimination and substance use, with the effect being stronger in men than in women [34]. Similarly, a longitudinal study on Hispanic adolescent immigrants from Los Angeles and Miami, who had resided in the United States for five years, found that ethnic discrimination had significant effects on increased alcohol consumption, which was consistent for men and women [35]. Similarly, a study of Indian and Pakistani migrants in Hong Kong examined whether the reasons for drinking to cope with stress and improve their situation had mediating effects on the relationship between perceived discrimination and hazardous alcohol consumption. Racial discrimination was found to be associated with hazardous alcohol consumption, and this relationship was entirely mediated by the reason for coping with the situation [36]. Additionally, a longitudinal study of African American adolescents found that experiences of discrimination were positively associated with substance use, but this association was mediated by school participation and relationships with peers who consume substances. This finding suggests that discrimination leads to the observed associations with substance use and not the other way around [37].

Despite the abundant evidence on the association between discrimination and substance use, especially from studies conducted in the United States, there is a lack of studies on this association in the context of Latin American adolescents, mainly in Chile, which has a high prevalence of alcohol and marijuana use in its population [15,38].

Although the Chilean government and its respective institutions (educational and public safety) have implemented various preventive programs to reduce and control substance use among adolescents, they are characterized by maintaining epidemiological monitoring, in addition to guiding educational institutions to incorporate specific protocols and norms. These are mostly related to developing activities that promote healthy behaviors, constant communication with students’ families, addressing specific issues of drug knowledge, resistance to social pressure, peer influence, mental health status, and access to marijuana and/or alcohol. These do not include the effects of everyday discrimination [39]. Therefore, considering the effects of discrimination and substance use on adolescents’ health and academic achievements [40,41,42], it is necessary to obtain information that can complement and contribute to the design of preventive programs in educational institutions.

This study examined the relationship between everyday discrimination and alcohol and marijuana use in Chilean adolescents. These findings will allow us to obtain new, novel information that could inform future interventions in schools in a country where mental health services still have serious inequalities in terms of access and where the prevalence of marijuana use, since 2011, has more than doubled [15,43].

## 2. Materials and Methods

### 2.1. Design and Participants

A non-experimental study with a cross-sectional predictive design was used because the purpose was to explore the functional relationship by predicting a criterion variable from one or more predictors, and all variables were measured at a single moment in time [44].

This study collected data using a non-probability sampling strategy based on availability [45]. A total of 2329 middle school (7–8 grades) and high school (9–12 grades) students from the city of Arica, Chile, participated. This equated to a response rate of 77.6% of the available sample size (*n* = 3000). Of the students, 45.8% (*n* = 1068) attended schools subsidized by the government, 41.8% (*n* = 972) attended public schools, and 12.4% (*n* = 289) attended private schools. These three types of educational institutions are differentiated according to the type of financing and administration: (1) public schools are administered and financed by the government, (2) subsidized schools have a private administration, and the financing is shared (government and parents); and (3) in private schools, the administration and financing are private. These differences have an impact on the quality of education that students can access, grouping families from different socioeconomic levels according to the type of educational institution [46,47,48]. It should be noted that access to the sample largely depended on the availability of time (slot between different extracurricular activities) and the willingness to participate on the part of educational institutions.

### 2.2. Instruments

The Sistema de evaluación de niños y adolescentes (SENA; Child and Adolescent Assessment System) [49,50] instrument was developed by specialists in psychopathology and psychological assessment. The purpose was to measure a wide range of emotional and behavioral problems among three age groups: infant (3–6 years), primary (6–12 years), and secondary (12–18 years). The response options for each version correspond to behavioral statements on a 5-point Likert scale (1 = “Never” to 5 = “Always”). Sánchez-Sánchez et al. [51] obtained adequate evidence of the validity and reliability (α > 0.70) for each subscale to be used for children and adolescents in Spanish-speaking contexts. For this study, only the self-report version of students aged 12 to 18 years regarding alcohol and marijuana consumption was used.

Substance Use (SUS): Assesses problematic substance use in adolescents, specifically alcohol and marijuana use, through six items. This questionnaire uses direct items (“When I am with my friends, I smoke marijuana or joints”) and indirect items (“My friends take drugs when they are with me”). Scores were obtained using the mean of all items. High scores indicate adolescent use, which could become problematic.

The Everyday Discrimination Scale (EDS) [52] is a brief nine-item instrument designed to measure routine experiences of discrimination associated with unfair treatment that occurs on a daily basis. The factorial structure is one-dimensional and contains items, such as “You are treated with less courtesy than other people” and “People act as if they are afraid of you”. The response options correspond to attitudinal/behavioral statements on a six-point Likert scale (1 = “almost every day” to 6 = “never”). Response options for the items were reversed. Higher scores suggest higher levels of everyday discrimination. Krieger et al. [53] translated and adapted this scale into Spanish, reporting adequate levels of reliability and construct validity. This scale has also shown adequate evidence of validity (able to explain 49% of the variance) and reliability (α = 0.87) in a sample of adolescents [54].

Sociodemographic data were collected using an ad hoc scale that included questions on gender, age, nationality, ethnicity, and vulnerability indices of the students. The vulnerability indices are composed of a high and low vulnerability classification based on the statistics presented in the annual municipal development plan of Arica [55]. From 2014 to 2017, vulnerability indices for municipal schools averaged 86%, while government-subsidized and private schools averaged 74% [56]. The communal vulnerability percentage (77%) was used as the cut-off point. Students who belonged to schools above the average vulnerability percentage of the commune were classified as “highly vulnerable”, and those who belonged to schools below the average vulnerability percentage of the commune were classified as “low vulnerability”.

### 2.3. Procedure

The Scientific Ethical Committee of the University of Tarapacá (no. 26.2017) approved this study.

The directors of the educational institutions were contacted and invited to participate voluntarily in this study. Subsequently, informed consent was obtained from the parents at the parent-teacher meetings. Once the parents’ authorization was obtained, we proceeded to obtain the consent of the adolescents, and then we began data collection using a paper and pencil instrument that was completed in a group setting inside one of their classes between March and December 2018. In each class, there were at least two psychologists in charge of the evaluation, who explained to the students the confidentiality of their answers and how the data would be used. The students did not receive any financial incentive for participation. The completion of the questionnaire required approximately 45 min.

### 2.4. Statistical Analysis

Initially, two procedures were used to process the missing data: (1) the replacement of the missing value with the median on the scale and (2) reporting as missing values. The first procedure was used in all instruments with missing values below 3%, whereas the second was only used in the everyday discrimination scale (EDS), which had between 10% and 15.9% missing data in its items. Therefore, when calculating the mean score, only 45 (1.9%) cases lost all item information; that is, the total mean EDS score was obtained with 2284 cases.

For characterizing the sample, proportions were calculated for each categorical variable, and the mean, standard deviation, minimum and maximum, skewness, kurtosis, and Shapiro–Wilk normality test [57] were obtained for each continuous variable. Furthermore, evidence of the preliminary validity and reliability of the scales was obtained for this sample, in order to ensure appropriate interpretation of the scores in this study. Validity evidence based on the internal structure of the test was obtained through confirmatory factor analysis (CFA), with a WLSWV estimation method, which is robust with non-normal discrete variables [58,59] and from the polychoric correlation matrix, given the ordinal structure of the data [60]. Fit was assessed following the cut-point recommendations proposed by Schreiber in 2017 [61] for the comparative fit index (CFI), Tucker–Lewis index (TLI), and root mean squared error of approximation (RMSEA; for example, CFI > 0.95, TLI > 0.95, RMSEA < 0.06). The reliability was estimated from Cronbach’s alpha and McDonald’s omega coefficients, both in their non-ordinal versions [62].

Comparisons were made on the basis of the mean scores of the substance use variable by gender (men and women), age group (11–13 years and 14–19 years), ethnicity (Aymara and non-Aymara students), and vulnerability index (high and low), using the independent samples *t*-test. Since the data did not present homoscedasticity in the substance use variable and the grouping variables (gender: Levene’s test, F = 19.133, *p* < 0.001; age group: Levene’s test, F = 425.731, *p* < 0.001; ethnicity: Levene’s test, F = 8.088, *p* = 0.004; vulnerability index: Levene’s test, F = 17.376, *p* < 0.001), Welch’s *t*-test was used. To assess the effect size of the differences, the coefficient d proposed by Cohen [63] was estimated. Although the variables in this sample were not normally distributed, parametric comparative analyses were used, as the t-statistic is sufficiently robust under conditions of skewness and with large sample sizes (*n* > 50) [64,65].

The association between substance use and everyday discrimination scores was estimated using Pearson’s correlation coefficient. Subsequently, to assess the predictive capacity of everyday discrimination on substance use (criterion variable), a multiple linear regression analysis was performed. To control for the effects of sociodemographic variables, two blocks were formed. In the first, the variables gender, age, ethnicity, and vulnerability index were entered. In the second, the everyday discrimination variable was entered. A stepwise method was used to determine the variables in each block. The final model incorporated standardized β coefficients, which represent the changes in the standard deviation of the criterion variable. The predictive variables with the largest beta standardized coefficients suggest a greater relative effect on substance use. All assumptions were met, except for normality. The presence of collinearity between the independent variables was discarded by means of the inflated variance factor, which was less than two for all of them. The residuals were independent of each other (Durbin–Watson statistic = 1.915). Homoscedasticity was confirmed through a dispersion plot of predictors and residuals. Normality was discarded through the Q–Q plot standardized residuals. The statistical hypothesis testing of the data analyses was performed at a 5% significance level; thus, the conditional probability of a type I error was 5% if the null hypothesis was considered true.

All statistical analyses were performed using the statistical software IBM SPSS (version 25) [66], Mplus (version 8.2) [67], and JASP (version 0.14.1) [68].

## 3. Results

Study participants included 2329 students. Their age ranged from 11 to 19 years, and the mean age was 14.3 years (*SD* = 1.8). Total 1170 (50.2%) were female, 1357 (58.3%) had low vulnerability, 1651 (70.9%) were non-Aymara, and 2166 (93%) were Chilean. Table 1 presents the sociodemographic details.

Participants had mean scores close to the lower level on substance use (*M* = 1.23; *SD* = 0.52; *Min–Max* = 1.0. 5.0) and everyday discrimination (*M* = 2.14; *SD* = 1.02; *Min–Max* = 1.0–6.0), showing that most responses focused on reporting a low frequency of substance use and everyday discrimination. The standardized skewness and kurtosis ratios were outside the recommended range [64] (substance use, *S* = 59.7, *K* = 104.6; everyday discrimination, *S* = 20.4, *K* = 6.9). This suggests that the variables had a positively skewed leptokurtic distribution. The Shapiro-Wilk test showed that neither variable was normally distributed (substance use, F = 0.527, *p* = < 0.001; everyday discrimination, F = 0.904, *p* < 0.001).

The CFA results show that the fit of the substance use and everyday discrimination scales were in accordance with the standards recommended in the literature (CFI > 0.95; TLI > 0.95), with the exception of the RMSEA value, which exceeds the criterion of 0.06 (see SUS and EDS in Table 2) [61]. To determine whether the measurement models needed debugging or respect, the modification indices were reviewed. This suggested a correlation between the errors of two items, as this could increase the model fit. For substance use, items 74 and 55 (r = 0.305, *p* < 0.001) and 159 and 142 (r = 0.212, *p* < 0.001) were correlated. For everyday discrimination, items 1 and 2 (r = 0.249, *p* < 0.001) and 8 and 9 (r = 0.238, *p* < 0.001) were correlated. Once the measurement models were debugged, the fit improved significantly for both scales. Thus, they were adequate population representations of the observed covariate matrix. Reliability estimates were satisfactory (SUS, ω = 0.85, α = 0.83; EDS, ω = 0.88, α = 0.87) [69]. Table 2 shows the fit indices for the CFA details.

Welch’s *t*-test showed that there were differences between substance use by gender (men = 1.27 (*SD* = 0.55); women = 1.20 (*SD* = 0.49); *t* = −3.068; *p* = 0.002; d = −0.128), age group (11–13 years = 1.06 (*SD* = 0.22); 14–19 years = 1.34 (*SD* = 0.61); *t* = −15.680; *p* < 0.001; d = −0.608), ethnicity (Aymara = 1.19 (*SD* = 0.46); non-Aymara = 1.24 (*SD* = 0.54); *t* = −2.215 *p* = 0.027; d = −0.102), and vulnerability (low = 1.21 (*SD* = 0.47); high = 1.26 (*SD* = 0.58); *t* = 2.341; *p* = 0.019; d = 0.100). There were differences between small and moderate effect sizes [63]. This suggests that men who were non-Aymara, aged 14 to 19 years, and with high vulnerability might indulge in greater amount of substance use associated behaviors than their women counterparts.

Pearson’s correlation analyses showed that substance use had a weak statistically significant and direct correlation with everyday discrimination (r = 0.138; *p* < 0.001) [63]. In the multivariate analysis, substance use was considered a criterion variable. The first block included gender, age, ethnicity, and vulnerability index as predictor variables, while the second block included everyday discrimination as a predictor variable.

The linear regression model of the first block was statistically significant (F = 61.906; *p* < 0.001). Gender, age, and ethnicity were entered into the regression equation, which explained 10% of the variability in substance use. It should be noted that the effects of gender and ethnicity, although statistically significant, were null. The linear regression model in the second block was also statistically significant (F = 59.446; *p* < 0.001). Age, ethnicity, and everyday discrimination entered the regression equation, explaining 11.7% of the variability in substance use. Although ethnicity had a statistically significant effect, it was null. This suggests that the closer students are to the legal age of the majority (18 years old in Chile) and experience higher levels of everyday discrimination, the higher the substance use they may exhibit. Table 3 presents the details of the multivariate analyses.

## 4. Discussions

The aim of this study was to describe the relationship between everyday discrimination and alcohol and marijuana consumption behaviors in a sample of adolescents in Chile. This study showed that everyday discrimination, together with the age of the adolescent, has direct effects on the consumption of alcohol and marijuana. These results are in accordance with those of previous studies that evaluated the relationship between discrimination and substance use, considering this association, both in terms of minority groups and the general population [27,28,34,35,36,37,70,71].

The increase in substance use with age could be explained by several psychosocial and environmental factors, including the increasing salience of the need for peer approval, the increased opportunities for substance consumption (parties, concerts, etc.), and the increasing need to distance themselves from their parents as they age. It is also possible that there are biological changes that increase vulnerability to substance use as patterns of use become habitual [17,72,73].

On the other hand, the effects of everyday discrimination on substance use could be explained by the value that adolescents assign to its consumption, specifically those belonging to the majority group, since it is possible that in an attempt to establish links and achieve greater integration, students belonging to the discriminated group increase their consumption of alcohol and/or marijuana, transforming substance use into a way of avoiding discrimination. Alternatively, discrimination is a source of stress, and research reveals that many victims of discrimination turn to the use of substances to cope with and find relief from the stress of discrimination. Multiple studies have documented a positive association between discrimination and substance use [74]. Therefore, to confront discrimination, many young people may become involved in greater ingestion of alcohol and/or marijuana to ameliorate the negative emotions triggered by experiences of discrimination directed toward them by the majority group. Discriminatory experiences can become an important risk factor in substance use, especially if these experiences are embedded in the daily and educational structures surrounding the adolescents. It should be considered that there are a number of interventions in schools that have been able to reduce stereotypes and discriminatory tendencies toward other groups [75] and even increasing positive feelings and willingness to engage in social contact with other students from other ethnic groups. For instance, the *Extended Class Exchange Program (ECEP)* [76], aiming to improve school climate, prosocial peer norms, confidence, and self-efficacy to address racism among students with “Speak Out Against Racism” [77].

According to the results obtained in this country, it is evident that educational institutions, through national preventive programs or their own initiatives, focus their efforts on designing and implementing psychosocial and/or psychoeducational interventions that include topics on the effects of discrimination and substance use. This is because, in recent years, they have not been considered within the “Choose to live without drugs” (*“Elige vivir sin drogas”*) program conducted by the National Service for the Prevention and Rehabilitation of Drug and Alcohol Consumption (SENDA), which, despite having a multidimensional and broad focus, could be hiding the role of discrimination in the consumption of alcohol and marijuana. In this sense, school counselors should also coordinate support networks to identify and support students who feel discriminated against, with the aim of achieving lower levels of substance use or delaying its onset.

This study has some limitations. First, a self-report measure was used to assess everyday discrimination, which, according to Parker [78], may result in inconsistencies when measuring discrimination. This is because people may perceive a negative result as discrimination, while others may perceive it as a result of bad luck or some other factor. Additionally, some people may cope with acts of discrimination by denying them or not reporting them because they feel ashamed or uncomfortable [79]. At the same time, some evidence suggests that the majority of respondents have a reasonably accurate recall of prior stressful events [79]. The second limitation is that this study did not contain reports from parents or teachers. A third limitation is related to a possible bias in terms of data collection; due to the fact that the assessment was carried out within the same classroom, it is possible that there was a certain degree of peer pressure. Fourth, in the debugging of the measurement models, to improve the fit of substance use and everyday discrimination scales, modification indices were used to allow two errors to correlate with each other. This strategy is justified by the overlapping meanings or contents of the items. The final limitation corresponds to the study design. Since this study was cross-sectional, we have no information about the temporal ordering among the variables assessed, and no causal effects can be established among them.

Future research could examine whether the association between discrimination and substance use is stable over time or if it varies at different stages of consumption (experimental use and more regular and active use). Research should also evaluate the potential mediating or moderating effects of the influence of family, school, and peers on the relationship between discrimination and substance use. Prior research does indicate that social ties can reduce the negative effects of discrimination on health [80].

## 5. Conclusions

The results of this study show that daily discrimination and adolescent age are positively associated with substance use, suggesting that as adolescents age and/or witness more experiences of discrimination, they are likely to consume more alcohol and marijuana. These results have implications at the individual level. The need to address and improve coping strategies in children and adolescents to deal with possible experiences of discrimination is observed at the school level, which is a space where interventions that allow addressing discrimination should be facilitated. There are also public policy implications where constant monitoring of the effects of ongoing cultural and policy changes on marijuana use is applied [15] and also in terms of the increase in alcohol consumption.

Further research in this area should be considered a priority for future research among adolescents in Chile.

## Figures and Tables

**Table 1 ijerph-18-06485-t001:** Sociodemographic characteristics (*n* = 2329).

Variable	Sociodemographic Data	*n* (%)
Gender	Men	1155 (49.6%)
	Women	1170 (50.2%)
	Missing	4 (0.2%)
Age (dichotomized)	11–13	888 (38.1%)
	14–19	1435 (61.6%)
	Missing	6 (0.3%)
Course	Seventh grade	510 (21.9%)
	Eighth grade	461 (19.8%)
	Ninth grade	406 (17.4%)
	Tenth grade	358 (15.4%)
	Eleventh grade	328 (14.1%)
	Twelfth grade	266 (11.4%)
Vulnerability	Low	1357 (58.3%)
	High	972 (41.7%)
Nationality	Chilean	2166 (93.0%)
	Foreign	155 (6.7%)
	Missing	8 (0.3%)
Ethnicity	Aymara	606 (26.0%)
	Non-Aymara	1651 (70.9%)
	Missing	72 (3.1%)

*n* = Number of individuals; % = effective (percentage).

**Table 2 ijerph-18-06485-t002:** Fit indexes for CFA.

Scales	Par	*X* ^2^	*df*	RMSEA	90%CI	CFI	TLI	SRMR
SUS	35	481.047	9	0.191	[0.177–0.206]	0.958	0.930	0.047
SUSdebugged	37	54.305	7	0.69	[0.052–0.086]	0.996	0.991	0.016
EDS	61	684.183	27	0.131	[0.123–0.140]	0.952	0.936	0.042
EDSdebugged	63	157.198	25	0.061	[0.052–0.071]	0.990	0.986	0.023

Note: Par = parameters; χ^2^ = chi-square; *df* = degree of freedom; RMSEA = root mean square error of approximation; 90% CI = 90% confidence interval; CFI = comparative fit index; TLI = Tucker-Lewis index; SRMR = standardized root mean square residual.

**Table 3 ijerph-18-06485-t003:** Factors associated with substance use.

Block	Variable	Multivariate Analysis			
		*ß*	*SE*	*ß* Standardized	*p*-Value
1	Intercept	−0.166	0.105	–	–
	Gender	0.048	0.021	0.047	**0.021**
	Age in years	0.088	0.006	0.307	**<0.001**
	Ethnicity	0.056	0.024	0.048	**0.020**
	Vulnerability	−0.018	0.022	−0.017	0.419
	R-square corrected	0.100			
2	Intercept	−0.289	0.105	–	–
	Gender	0.037	0.021	0.036	0.075
	Age in years	0.087	0.006	0.306	**<0.001**
	Ethnicity	0.055	0.024	0.047	**0.021**
	Vulnerability	−0.017	0.022	−0.016	0.426
	Everyday discrimination	0.068	0.010	0.134	**<0.001**
	R-square corrected	0.117			

*ß* = beta coefficient; *SE* = standard error; *ß* standardized = standardized beta coefficient; values in bold indicate statistical significance.

## Data Availability

The data supporting the results of this article will be made available by the authors, without undue reservation, to any qualified researcher.

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
