# Peer review of "Relationship between Everyday Discrimination and Substance Use among Adolescents in Northern Chile"

_ijerph, 2021, doi:10.3390/ijerph18126485_

Round 1
Reviewer 1 Report
IJERPH 1252935 v1. Peer Review. 31st of May 2021.
The study by Caqueo-Urizar et al is investigating the possible relationship between discrimination and substance use in a cohort of adolescents (12-20 years old) from Northern Chile (Arica). Using a cross-sectional approach, the authors focused on alcohol and/or marijuana use linked to discrimination. The results described herein show a positive association between discrimination, substance use and age.
This study is interesting and will be of value to the current literature. While the manuscript is well written, easy to read and easy to follow, some editing needs to be performed before publication. These minor points are detailed below, point-by-point.
It is rather strange that this peer review manuscript is not showing line numbers. It is therefore difficult to refer to specific sentences without having line numbers.
Introduction. 1st paragraph. Please edit the sentence beginning with “And”, as a sentence cannot begin with “and”. Either remove the ‘and’ or merge this sentence with the previous one (‘research suggest that young people…’).
Introduction. 2nd paragraph. Typographical error. Please edit the location of references [9,10], as they appear after a full stop.
Introduction. 5th paragraph. Typographical error. Please insert a space between “personality traits” and references number [15,16].
Introduction. 6th paragraph. Typographical error. Please insert a space between “according to Jones” and reference number [26].
Introduction. 6th paragraph. Typographical error. Please insert a space between “to the majority group” and reference number [30].
Introduction. 6th paragraph. Regarding the sentence “In a similar vein, a longitudinal study of Hispanic adolescents…”, could the authors explicit if the study was also (as in previous sentence) performed in the United States ? If not, the country in which this study was conducted should be explicitly stated. Besides, at the end of the last sentence, a space should be inserted between “females” and reference number [33]. Additionally, authors should rather use “men” and “women” instead of ‘males’ and ‘females’, since this study has been performed on participants, not animals.
Introduction. Last paragraph. Typographical errors. Please insert a space between ‘cannabis’ and reference number [36]. Also, please insert a space between ‘adolescents’ and references number [37-39].
Introduction. Last paragraph. Please give sources/references regarding the information in “this association in Latin American adolescents is worrying, mainly in Chile, which has a high prevalence of alcohol and cannabis use in this population.”
Methods. Section 2.1. Typographical error. Please insert a space between “education institution” and references number [42, 43, 44]. Additionally, should these reference be [42-44] ?
Methods. Section 2.2. Typographical error. Please insert a space between “Assessment System” and references number [45-46].
Methods. Section 2.2. The sentence “For this study, only used the self-reported” needs to be edit, as it makes no sense at present, due to grammatical error.
Results. 2nd paragraph. The sentence “Pearson’s correlation analyses… have statistically significant and direct correlated with …” needs editing. I suggest using “correlation”.
Results. 2nd paragraph. Excessive parenthesis use in the sentence “The correlation was weak … “. Please edit unnecessary parenthesis, e.g. “(r < 0.1 )) [54] for gender”.
Discussion. 4th paragraph. Excessive use of “Thus” in e.g. “Thus, in order to confront … by the majority group. Thus, discriminatory experiences…”.
Discussion. Limitation paragraph. Authors should also include a possible bias regarding data collection, as it was performed under peer scrutiny. Indeed, in the methods, authors mentioned that “we began data collection using a paper and pencil instrument that was completed in a group setting inside one of their classes”. Therefore, if students completed the questionnaires while others did too, some peer pressure could have occurred, which could represent a (small) bias. This should, at least, be mentioned in the current limitation section.
Throughout the manuscript, authors use "cannabis" and "marijuana". While this is not false per se, there is a difference, especially when focusing on pharmacology, between "cannabis" and "marijuana". Authors should explain to what they are reffering.
While "cannabis" refers to many plants, "marijuana" rather refers to products with THC and/or CBD.
Reviewer 2 Report
All points of this study are presented clearly as regarding the relationship of everyday discrimination and substance use and limitation of the study is being set as well.
Clarify the weak correlation in the results.
Check the appropriate term like maladpatative or maladaptive.
Reviewer 3 Report
The material is interesting and the topic is relevant. The method seems to have been followed faithfully and the authors were well-positioned to conduct the analysis. Despite these positives in my view the paper needs more work before it could be published and I have made some specific suggestions below.
- The literature addressed is described accurately so far as I can see. Relevant literature is presented in support of the research problem. Further, there is no clear distinction between manuscript sections in terms of the content they report. First, I suggest dividing the section "Background" into three components, respectively introduction (explain the general argument of the paper, without going into specific details) background (situate the study concepts within the context of extant knowledge, discuss the international relevance of the concepts) and purpose, creating greater clarity in the analysis of the reader. What is the study's biggest contribution, is it sample? The contribution should be clearly stated in the introduction.
- The ethical aspects in collecting data are not specifically clarified, independently of the voluntary nature of the subjects´ participation and the approval by the local IRB; variables such as the offer of incentives to participate (how participants were compensated for participation), sharing and use of data are not patent. More information is needed about the issues around informed consent or confidentiality or how they have handled the effects of the study on the participants during and after the study.
- There is no mention of the sample size that was targeted and obtained to meet the sample size requirements for data analysis. More precision is necessary regarding the sampling strategy and access to the target population. Response rate? How were participants recruited? When the data were collected? Data collection period?
- A better visual structure of tables (boldface variables with statistical significance) would improve the readability.
- The discussion section should be reorganized because they are poor. I believe there should be a better integration of the results with the existing literature.
- In the conclusion section, there is a complete absence of the empirical implications of the study, besides which the theoretical implications should have been approached in greater depth; Also implications for practice and research need addressed in more deep. I suggest dividing this theoretical implications/ recommendations for action, in three ways: - individual actions; - educational/school responsibilities; and policy implications.
CHECKLIST FOR STYLE.
- The manuscript will serve a broad audience of students, researchers, and practitioners, however the manuscript needs to be carefully and attentively proofread, because some sentences are awkwardly constructed, punctuation is deficient, and therefore reading is occasionally difficult to follow. Would recommend a thorough technical edit of this paper.
Reviewer 4 Report
"Relationship between everyday discrimination and substance use among adolescents in Northern Chile"
Thank you for the opportunity to read this paper on the relationship between everyday discrimination and substance use and its determinants.
- Introduction needs a thorough overhaul. In my view, you must introduce and delineate the key concepts, you are working with. Focus on specifying your key concept, research aims, objectives and design. However, I found invalid knowledge contributions to investigate between discrimination and substance use. Moreover, the introduction is very general highlight how and why is needed testable the relationship.
- Methods are rather weak and intransparent. This concerns sampling and sample (unclear and not well anchored) as well as a measure (unclear), data collection (unclear) and data analysis (inconsistent and insufficiently justified).
- The findings are largely invalid and missing. Using wrong statistics for testing the relationship. For instance, SUS-6 items and EDS-9 items, my question is, these items are really represented of theoretical measurement. I found reliability is very high. In table 3, you test the relationship, but substance use is missing.
- Discussion is missing an overarching theory. Lack of clear what main findings contributed to new argument.
- The conclusion is very short. It is nor clear, how can use to be practical implications.
- Reference should be correct. Many text and reference list are missing and out-up-date.
- The language used does not reflect an academic journal. Many grammar errors.
Reviewer 5 Report
This cross-sectional study aimed at evaluating the relationship between everyday discrimination and alcohol and cannabis use in a sample including 2330 Chilean adolescents. The authors observed a significant association between age of participants as well as everyday discrimination and use of these substances. The topic is relevant and the results are of high interest. The article is well written. Following are a few comments on aspects that might be improved.
- In the first sentence of the introduction, please replace "the use of substances, such as alcohol and cannabis are considered ... " with "the use of substances, such as alcohol and cannabis is considered ... "
- In the Statistical analysis section, please report whether a test was used to assess normality of distribution and if parametric or non-parametric tests were chosen accordingly
- In the same section, please report whether assumptions to construct a linear regression model (besides collinearity which is reported) were evaluated and in which way
- In the model in which age of participants and everyday discrimination were associated with substance use scores, was there a significant interaction between these two terms? This could be tested in order to understand whether everyday discrimination exerts a higher effect in one of the two age groups that were defined by the authors
Round 2
Reviewer 3 Report
Thank you for the opportunity to contribute to the improvement of your manuscript. Overall, your manuscript has improved with the recommendations of the various reviewers. Congratulations.
Reviewer 4 Report
Overall, the revised version is read well than the first time. All comments are revised.